# Improving Computational Efficiency in Visual Reinforcement Learning via Stored Embeddings

Lili Chen[1]    Kimin Lee[1]    Aravind Srinivas[2]    Pieter Abbeel[1]

[1]UC Berkeley   [2]OpenAI

## Abstract

Recent advances in off-policy deep reinforcement learning (RL) have led to impressive success in complex tasks from visual observations. Experience replay improves sample-efficiency by reusing experiences from the past, and convolutional neural networks (CNNs) process high-dimensional inputs effectively. However, such techniques demand high memory and computational bandwidth. In this paper, we present Stored Embeddings for Efficient Reinforcement Learning (SEER), a simple modification of existing off-policy RL methods, to address these computational and memory requirements. To reduce the computational overhead of gradient updates in CNNs, we freeze the lower layers of CNN encoders early in training due to early convergence of their parameters. Additionally, we reduce memory requirements by storing the low-dimensional latent vectors for experience replay instead of high-dimensional images, enabling an adaptive increase in the replay buffer capacity, a useful technique in constrained-memory settings. In our experiments, we show that SEER does not degrade the performance of RL agents while significantly saving computation and memory across a diverse set of DeepMind Control environments and Atari games.

## 1   Introduction

Success stories of deep reinforcement learning (RL) from high dimensional inputs such as pixels or large spatial layouts include achieving superhuman performance on Atari games [30, 37, 1], grandmaster level in Starcraft II [50] and grasping a diverse set of objects with impressive success rates and generalization with robots in the real world [21]. Modern off-policy RL algorithms [30, 15, 11, 12, 39, 22, 24] have improved the sample-efficiency of agents that process high-dimensional pixel inputs with convolutional neural networks (CNNs; LeCun et al. 25) using past experiential data that is typically stored as raw observations in a replay buffer [28]. However, these methods demand high memory and computational bandwidth, which makes deep RL inaccessible in several scenarios, such as learning with much lighter on-device computation (e.g. mobile phones or other light-weight edge devices).

For compute- and memory-efficient deep learning, several strategies, such as network pruning [13, 8], quantization [13, 17] and freezing [53, 36] have been proposed in supervised learning and unsupervised learning for various purposes (see Section 2 for more details). In computer vision, Raghu et al. [36] and Brock et al. [5] showed that the computational cost of updating CNNs can be reduced by freezing lower layers earlier in training, and Han et al. [13] introduced a deep compression, which reduces the memory requirement of neural networks by producing a sparse network. In natural language processing, several approaches [46, 42] have studied improving the computational efficiency of Transformers [49]. In deep RL, however, developing compute- and memory-efficient techniques has received relatively little attention despite their serious impact on the practicality of RL algorithms.

In this paper, we propose **S**tored **E**mbeddings for **E**fficient **R**einforcement Learning (SEER), a simple technique to reduce computational overhead and memory requirements that is compatible with various

35th Conference on Neural Information Processing Systems (NeurIPS 2021).

off-policy RL algorithms [10, 15, 39]. Our main idea is to freeze the lower layers of CNN encoders of RL agents early in training, which enables two key capabilities: (a) compute-efficiency: reducing the computational overhead of gradient updates in CNNs; (b) memory-efficiency: saving memory by storing the low-dimensional latent vectors to experience replay instead of high-dimensional images. Additionally, we leverage the memory-efficiency of SEER to adaptively increase replay capacity, resulting in improved sample-efficiency of off-policy RL algorithms in constrained-memory settings. SEER achieves these improvements without sacrificing performance due to early convergence of CNN encoders.

The main contributions of this paper are as follows:

- We present SEER, a compute- and memory-efficient technique that can be used in conjunction with most modern off-policy RL algorithms [10, 15].

- We show that SEER significantly reduces computation while matching the original performance of existing RL algorithms on both continuous control tasks from DeepMind Control Suite [45] and discrete control tasks from Atari games [2].

- We show that SEER improves the sample-efficiency of RL agents in constrained-memory settings by enabling an increased replay buffer capacity.

## 2 Related work

**Off-policy deep reinforcement learning.** The most sample-efficient RL agents often use off-policy RL algorithms, a recipe for improving the agent's policy from experiences that may have been recorded with a different policy [44]. Off-policy RL algorithms are typically based on Q-Learning [51] which estimates the optimal value functions for the task at hand, while actor-critic based off-policy methods [27, 38, 10] are also commonly used. In this paper we will consider Deep Q-Networks (DQN; Mnih et al. 30),which combine the function approximation capability of deep convolutional neural networks (CNNs; LeCun et al. 25) with Q-Learning along with the usage of the experience replay buffer [28] as well as off-policy actor-critic methods [27, 10], which have been proposed for continuous control tasks.

Taking into account the learning ability of humans and practical limitations of wall clock time for deploying RL algorithms in the real world, particularly those that learn from raw high dimensional inputs such as pixels [21], the sample-inefficiency of off-policy RL algorithms has been a research topic of wide interest and importance [23, 20]. To address this, several improvements in pixel-based off-policy RL have been proposed recently: algorithmic improvements such as Rainbow [15] and its data-efficient versions [48]; using ensemble approaches based on bootstrapping [34, 26]; combining RL algorithms with auxiliary predictive, reconstruction and contrastive losses [18, 16, 33, 52, 39, 40]; using world models for auxiliary losses and/or synthetic rollouts [43, 9, 20, 12]; using data-augmentations on images [24, 22].

**Compute-efficient techniques in machine learning.** Most recent progress in deep learning and RL has relied heavily on the increased access to more powerful computational resources. To address this, Mattson et al. [29] presented MLPerf, a fair and precise ML benchmark to evaluate model training time on standard datasets, driving scalability alongside performance, following a recent focus on mitigating the computational cost of training ML models. Several techniques, such as pruning and quantization [13, 8, 4, 17, 46] have been developed to address compute and memory requirements. Raghu et al. [36] and Brock et al. [5] proposed freezing earlier layers to remove computationally expensive backward passes in supervised learning tasks, motivated by the bottom-up convergence of neural networks. This intuition was further extended to recurrent neural networks [31] and continual learning [35], and Yosinski et al. [53] study the transferability of frozen and fine-tuned CNN parameters. Fang et al. [7] store low-dimensional embeddings of input observations in scene memory for long-horizon tasks. We focus on the feasibility of freezing neural network layers in deep RL and show that this idea can improve the compute- and memory-efficiency of many off-policy algorithms using standard RL benchmarks.

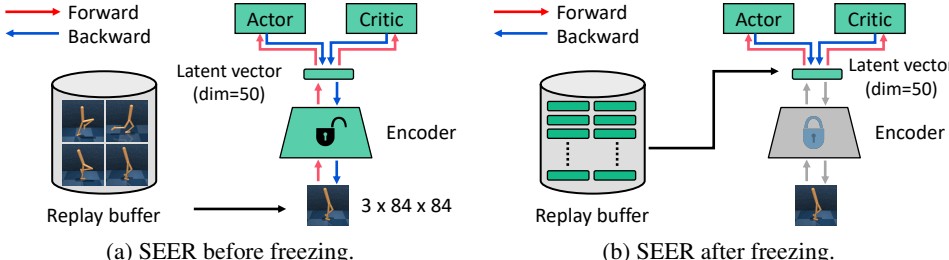

(a) SEER before freezing. (b) SEER after freezing.

Figure 1: Illustration of our framework. (a) Before the encoder is frozen, all forward and backward passes are active through the network, and we store images in the replay buffer. (b) After freezing, we store latent vectors in the replay buffer, and remove all forward and backward passes through the encoder. We remark that more samples can be stored in the replay buffer due to the relatively low dimensionality of the latent vector.

## 3 Background

We formulate visual control task as a partially observable Markov decision process (POMDP; Sutton & Barto 44, Kaelbling et al. 19). Formally, at each timestep $t$, the agent receives a high-dimensional observation $o_t$, which is an indirect representation of the state $s_t$, and chooses an action $a_t$ based on its policy $\pi$. The environment returns a reward $r_t$ and the agent transitions to the next observation $o_{t+1}$. The return $R_t = \sum_{k=0}^{\infty} \gamma^k r_{t+k}$ is the total accumulated rewards from timestep $t$ with a discount factor $\gamma \in [0, 1)$. The goal of RL is to learn a policy $\pi$ that maximizes the expected return over trajectories. By following the common practice in DQN [30], we handle the partial observability of environment using stacked input observations, which are processed through the convolutional layers of an encoder $f_\psi$.

**Soft Actor-Critic**. SAC [10] is an off-policy actor-critic method based on the maximum entropy RL framework [55], which encourages robustness to noise and exploration by maximizing a weighted objective of the reward and the policy entropy. To update the parameters, SAC alternates between a soft policy evaluation and a soft policy improvement. At the soft policy evaluation step, a soft Q-function, which is modeled as a neural network with parameters $\theta$, is updated by minimizing the following soft Bellman residual:

$$\mathcal{L}_Q^{\text{SAC}}(\theta, \psi) = \mathbb{E}_{\tau_t \sim \mathcal{B}}\Bigg[\bigg(Q_\theta(f_\psi(o_t), a_t) - r_t$$
$$- \gamma \mathbb{E}_{a_{t+1} \sim \pi_\phi}\big[Q_{\bar{\theta}}(f_{\bar{\psi}}(o_{t+1}), a_{t+1}) - \alpha \log \pi_\phi(a_{t+1}|f_\psi(o_{t+1}))\big]\bigg)^2\Bigg],$$

where $\tau_t = (o_t, a_t, r_t, o_{t+1})$ is a transition, $\mathcal{B}$ is a replay buffer, $\bar{\theta}, \bar{\psi}$ are the delayed parameters, and $\alpha$ is a temperature parameter. At the soft policy improvement step, the policy $\pi$ with its parameter $\phi$ is updated by minimizing the following objective:

$$\mathcal{L}_\pi^{\text{SAC}}(\phi) = \mathbb{E}_{o_t \sim \mathcal{B}, a_t \sim \pi_\phi}\big[\alpha \log \pi_\phi(a_t|f_\psi(o_t)) - Q_\theta(f_\psi(o_t), a_t)\big].$$

Here, the policy is modeled as a Gaussian with mean and covariance given by neural networks.

**Deep Q-learning.** DQN algorithm [30] learns a Q-function, which is modeled as a neural network with parameters $\theta$, by minimizing the following Bellman residual:

$$\mathcal{L}^{\text{DQN}}(\theta, \psi) = \mathbb{E}_{\tau_t \sim \mathcal{B}}\Bigg[\bigg(Q_\theta(f_\psi(o_t), a_t) - r_t - \gamma \max_a Q_{\bar{\theta}}(f_{\bar{\psi}}(o_{t+1}), a)\bigg)^2\Bigg],$$

where $\tau_t = (o_t, a_t, r_t, o_{t+1})$ is a transition, $\mathcal{B}$ is a replay buffer, and $\bar{\theta}, \bar{\psi}$ are the delayed parameters. Rainbow DQN integrates several techniques, such as double Q-learning [47] and distributional DQN [3]. For exposition, we refer the reader to Hessel et al. [15] for more detailed explanations of Rainbow DQN.

# 4 SEER: Stored Embeddings for Efficient Reinforcement Learning

In this section, we present SEER: **S**tored **E**mbeddings for **E**fficient **R**einforcement Learning, which can be used in conjunction with most modern off-policy RL algorithms, such as SAC [10] and Rainbow DQN [15]. Our main idea is to freeze lower layers during training and only update higher layers, which eliminates the computational overhead of computing gradients and updating in lower layers. We additionally improve the memory-efficiency of off-policy RL algorithms by storing low-dimensional latent vectors in the replay buffer instead of high-dimensional pixel observations. See Figure 1 and Appendix **??** for more details of our method.

## 4.1 Freezing encoder for saving computation and memory

We process high-dimensional image input with an encoder $f_\psi$ to obtain $z_t = f_\psi(o_t)$, which is used as input for policy $\pi_\phi$ and Q-function $Q_\theta$ as described in Section 3. In off-policy RL, we store transitions $(o_t, a_t, o_{t+1}, r_t)$ in the replay buffer $\mathcal{B}$ to improve sample-efficiency by reusing experience from the past. However, processing high-dimensional image input $o_t$ is computationally expensive. To handle this issue, after $T_f$ updates, we freeze the parameters of encoder $\psi$, and only update the policy and Q-function. We remark that this simple technique can save computation without performance degradation because the encoder is modeled as deep convolutional neural networks, while a shallow MLP is used for policy and Q-function. Freezing lower layers of neural networks also has been investigated in supervised learning based on the observation that neural networks converge to their final representations *from the bottom-up*, i.e., lower layers converge very early in training [36]. For the first time, we show the feasibility and effectiveness of this idea in pixel-based reinforcement learning (see Figure 7a for supporting experimental results) and present solutions to its RL-specific implementation challenges.

Moreover, in order to save memory, we consider storing (compressed) latent vectors instead of high-dimensional image inputs. Specifically, each experience in $\mathcal{B}$ is replaced by the latent transition $(z_t, a_t, z_{t+1}, r_t)$, and the replay capacity is increased to $\widehat{C}$ (see Section 4.2 for more details). Thereafter, for each subsequent environment interaction, the latent vectors $z_t = f_\psi(o_t)$ and $z_{t+1} = f_\psi(o_{t+1})$ are computed prior to storing $(z_t, a_t, z_{t+1}, r_t)$ in $\mathcal{B}$. During agent updates, the sampled latent vectors are directly passed into the policy $\pi_\phi$ and Q-function $Q_\theta$, bypassing the encoder convolutional layers. Since the agent samples and trains with latent vectors after freezing, we only store the latent vectors and avoid the need to maintain large image observations in $\mathcal{B}$.

## 4.2 Additional techniques and details for SEER

**Data augmentations.** Recently, various data augmentations [39, 24, 22] have provided large gains in the sample-efficiency of RL from pixel observations. However, SEER precludes data augmentations because we store the latent vector instead of the raw pixel observation. We find that the absence of data augmentations could decrease sample-efficiency in some cases, e.g., when the capacity of $\mathcal{B}$ is small. To mitigate this issue, we perform $K$ number of different data augmentations for each input observation $o_t$ and store $K$ distinct latent vectors $\{z_t^k = f_\psi(\text{AUG}_k(o_t)) | k = 1 \cdots K\}$. We find empirically that $K = 4$ achieves competitive performance to standard RL algorithms in most cases.

**Increasing replay capacity.** By storing the latent vector in the replay buffer, we can adaptively increase the capacity (i.e., total number of transitions), which is determined by the size difference between the input pixel observations and the latent vectors output by the encoder, with a few additional considerations. The new capacity of the replay buffer is

$$\widehat{C} = \left\lfloor C * \left( \tfrac{P}{4NKL} \right) \right\rfloor,$$

where $C$ is the capacity of the original replay buffer, $P$ is the size of the raw observation, $L$ is the size of the latent vector, and $K$ is the number of data augmentations. The number of encoders $N$ is algorithm-specific and determines the number of distinct latent vectors encountered for each observation during training. For Q-learning algorithms $N = 1$, whereas for actor-critic algorithms $N = 2$ if the actor and critic each compute their own latent vectors. Some algorithms employ a target network for updating the Q-function [30, 10], but we use the same latent vectors for the online and target networks after freezing to avoid storing target latent vectors separately and find that tying their

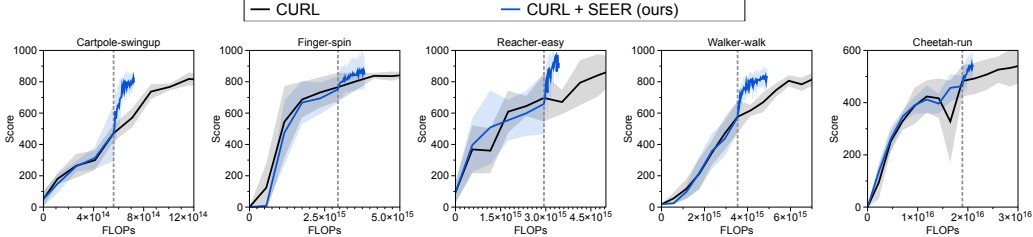

Figure 2: Learning curves for CURL with and without SEER, where the x-axis shows estimated cumulative FLOPs. The dotted gray line denotes the encoder freezing time $t = T_f$. The solid line and shaded regions represent the mean and standard deviation, respectively, across five runs.

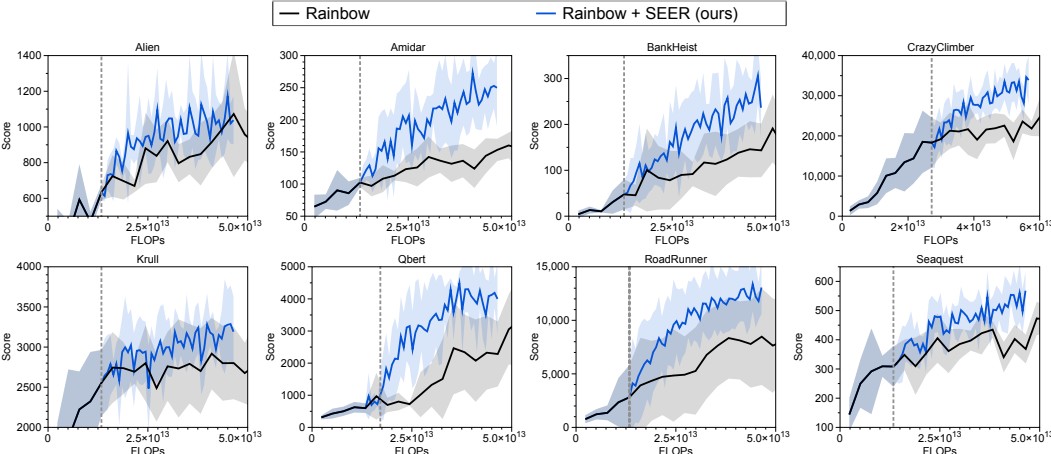

Figure 3: Learning curves for Rainbow with and without SEER, where the x-axis shows estimated cumulative FLOPs. The dotted gray line denotes the encoder freezing time $t = T_f$. The solid line and shaded regions represent the mean and standard deviation, respectively, across five runs.

parameters does not degrade performance.[1] The factor of 4 arises from the cost of saving floats for latent vectors, while raw pixel observations are saved as integer pixel values. We assume the memory required for actions and rewards is small and only consider only the memory used for observations.

## 5 Experimental results

We designed our experiments to answer the following questions:

- Can SEER reduce the computational overhead of various off-policy RL algorithms for both continuous (see Figure 2) and discrete (see Figure 3) control tasks?
- Can SEER reduce the memory consumption and improve the sample-efficiency of off-policy RL algorithms by adaptively increasing the buffer size (see Figure 4 and Figure 5)?
- Can SEER be useful for compute-efficient transfer learning (see Figure 7a)?
- Do CNN encoders of RL agents converge early in training (see Figure 8a and Figure 8b)?

### 5.1 Setups

**Compute-efficiency.** We first demonstrate the compute-efficiency of SEER on the DeepMind Control Suite (DMControl; Tassa et al. 45) and Atari games [2] benchmarks. DMControl is commonly used for benchmarking sample-efficiency for image-based continuous control methods. For DMControl experiments, we consider a state-of-the-art model-free RL method, which applies contrastive learning (CURL; Srinivas et al. 39) to SAC [10], using the image encoder architecture from SAC-AE [52].

---

[1]We remark that the higher layers of the target network are not tied to the online network after freezing.

|  | Scores at 45T FLOPs | | Scores at 500K environment steps (0.07GB) | |
| --- | --- | --- | --- | --- |
|  | Rainbow | Rainbow+SEER | Rainbow | Rainbow+SEER |
| Alien | $992.0 \pm 152.7$ | **1172.6** $\pm239.0$ | $1038.4 \pm 101.1$ | **1134.6** $\pm452.9$ |
| Amidar | $144.0 \pm 27.4$ | **250.5** $\pm47.4$ | $121.0 \pm 31.2$ | **165.3** $\pm47.6$ |
| BankHeist | $145.8 \pm 61.2$ | **276.6** $\pm98.1$ | **161.6** $\pm57.7$ | $151.8 \pm 65.8$ |
| CrazyClimber | $21580.0 \pm 3514.6$ | **28066.0** $\pm4108.5$ | $10498.0 \pm 1387.8$ | **17620.0** $\pm4418.4$ |
| Krull | $2799.5 \pm 468.1$ | **3277.5** $\pm440.5$ | $2215.7 \pm 336.9$ | **3069.2** $\pm377.6$ |
| Qbert | $2325.5 \pm 1152.7$ | **4123.5** $\pm1385.5$ | $2430.5 \pm 658.8$ | **3231.0** $\pm1567.6$ |
| RoadRunner | $10376.0 \pm 2886.0$ | **11794.0** $\pm1745.3$ | $10612.0 \pm 2059.3$ | **13064.0** $\pm2489.2$ |
| Seaquest | $402.8 \pm 48.4$ | **561.2** $\pm100.5$ | $262.8 \pm 19.1$ | **336.8** $\pm45.9$ |

Table 1: Scores on Atari games at 45T FLOPs corresponding to Figure 3 and at 500K environment interactions in the constrained-memory setup (0.07GB) corresponding to Figure 4. The results show the mean and standard deviation averaged five runs, and the best results are indicated in bold.

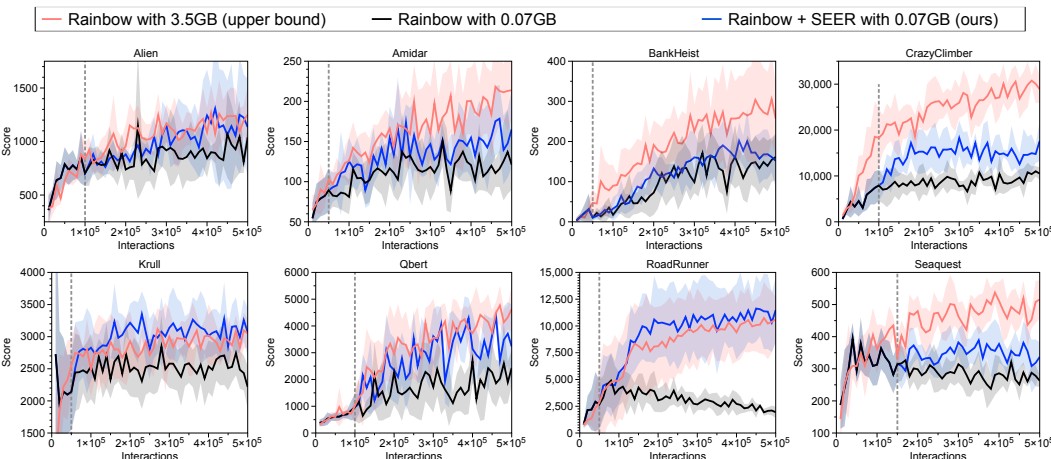

Figure 4: Comparison of the sample-efficiency of Rainbow with and without SEER in constrained-memory (0.07 GB) settings. The dotted gray line denotes the encoder freezing time $t = T_f$. The solid line and shaded regions represent the mean and standard deviation, respectively, across five runs.

For evaluation, we compare the computational efficiency of CURL with and without SEER by measuring floating point operations (FLOPs).[2]. For discrete control tasks from Atari games, we perform similar experiments comparing the FLOPs required by Rainbow [15] with and without SEER. For all experiments, we use the hyperparameters and architecture of data-efficient Rainbow [48].

**Memory efficiency.** We showcase the memory efficiency of SEER with a set of constrained-memory experiments in DMControl. For Cartpole and Finger, the memory allocated for storing observations is constrained to 0.03 GB, corresponding to an initial replay buffer capacity $C = 1000$. For Reacher and Walker, the memory is constrained to 0.06 GB for an initial capacity of $C = 2000$. In this constrained-memory setting, we compare the sample-efficiency of CURL with and without SEER. As an upper bound, we also report the performance of CURL without memory constraints, i.e., the replay capacity is set to the number of training steps. For Atari experiments, the baseline agent is data-efficient Rainbow and the memory allocation is 0.07 GB, corresponding to initial replay capacity $C = 10000$. The other hyperparameters are the same as those in the compute-efficiency experiments. Before the encoder is freeze, the replay buffer still needs to store the images and if the replay buffer slots number is equal with the baseline settings, the performance is equal to the baseline in theory. After the freeze time, the replay buffer slots number grows more larger. So the benefit is seems like on the condition of the assumption that a larger replay buffer would brings performance improvement? Such assumption needs to be claimed and discussed more clearly in the paper. Further discussions and experiments on the different limitations of the memory cost would be helpful.

---

[2]We explain our procedure for counting the number of FLOPs in Appendix **??**. The gain on wall-clock time is discussed in Appendix **??**.

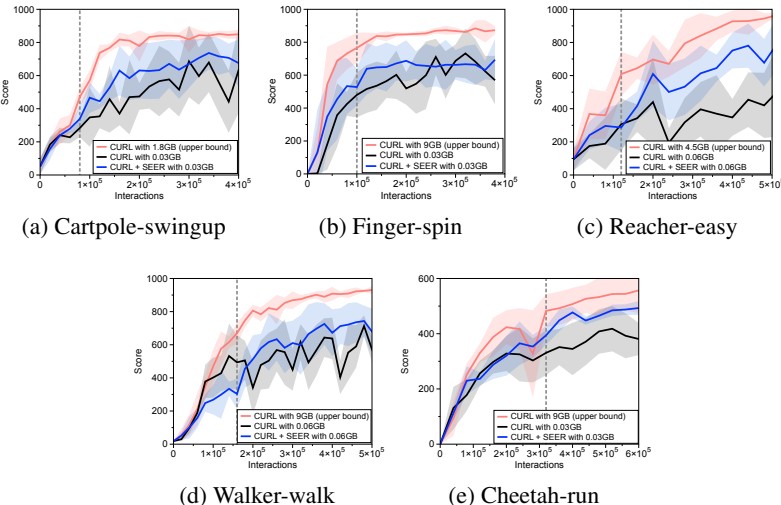

Figure 5: Comparison of the sample-efficiency of CURL with and without SEER in constrained-memory settings. The dotted gray line denotes the encoder freezing time $t = T_f$. The solid line and shaded regions represent the mean and standard deviation, respectively, across five runs.

The encoder architecture used for our experiments with CURL is used in Yarats et al. [52]. It consists of four convolutional layers with 3 x 3 kernels and 32 channels, with the ReLU activation applied after each conv layer. The architecture used for our Rainbow experiments is from van Hasselt et al. [48], consisting of a convolutional layer with 32 channels followed by a convolutional layer with 64 channels, both with 5 x 5 kernels and followed by a ReLU activation. For SEER, we freeze the first fully-connected layer in CURL experiments and the last convolutional layer of the encoder in Rainbow experiments. We present the best results across various values of the encoder freezing time $T_f$. See Appendices ?? and ?? for more hyperparameters and Appendix ?? for source code.

## 5.2 Improving compute- and memory-efficiency

Experimental results in DMControl and Atari showcasing the computational efficiency of SEER are provided in Figures 2 and Figure 3. CURL and Rainbow both achieve higher performance within significantly fewer FLOPs when combined with SEER in DMControl and Atari, respectively. Additionally, Table 1 compares the performance of Rainbow with and without SEER at 45T (4.5e13) FLOPs. In particular, the average returns are improved from 145.8 to 276.6 compared to baseline Rainbow in BankHeist and from 2325.5 to 4123.5 in Qbert. We remark that SEER achieves better computational efficiency while maintaining the agent's final performance and comparable sample-efficiency (see Appendix ?? for corresponding figures).

Experimental results in Atari and DMControl showcasing the sample-efficiency of SEER in the constrained-memory setup are provided in Figure 4 and Figure 5. CURL and Rainbow achieve higher final performance and better sample-efficiency when combined with SEER in DMControl and Atari, respectively. Additionally, Table 1 compares the performance of unbounded memory Rainbow and constrained-memory (0.07 GB) Rainbow with and without SEER at 500K interactions. In particular, the average returns are improved from 10498.0 to 17620.0 compared to baseline Rainbow in CrazyClimber and from 2430.5 to 3231.0 in Qbert. Although we disentangle the computational and memory benefits of SEER in these experiments, we also highlight the computational gain of SEER in constrained-memory settings (effectively combining the benefits) in Appendix ??. For an ablation on the freezing time, see Appendix ??. These experimental results show the real-world applicability of SEER (see Appendix ?? for more details).

## 5.3 Freezing larger convolutional encoders

We also verify the benefits of SEER using deeper convolutional encoders, which are widely used in a range of applications such as visual navigation tasks and favored for their superior generalization

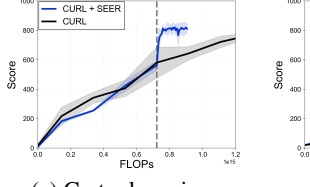
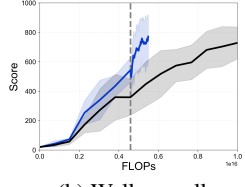

(a) Cartpole-swingup  (b) Walker-walk

Figure 6: Learning curves using IMPALA architecture, where the x-axis shows estimated cumulative FLOPs. The dotted gray line denotes the encoder freezing time $t = T_f$. The solid line and shaded regions represent the mean and standard deviation, respectively, across three runs.

ability. Specifically, we follow the setup described in Section 5.1 and replace the SAC-AE architecture (4 convolutional layers) with the IMPALA architecture [6] (15 convolutional layers containing residual blocks [14]). Figure 5.2 shows the computational efficiency of SEER in Cartpole-swingup and Walker-walk with the IMPALA architecture. CURL achieves higher performance within significantly fewer FLOPs when combined with SEER. We remark that the gains due to SEER are more significant because computing and updating gradients for large convolutional encoders is very computationally expensive.

## 5.4 Improving compute-efficiency in transfer settings

We demonstrate, as another application of our method, that SEER increases compute-efficiency in the transfer setting: utilizing the parameters from Task A on unseen Tasks B. Specifically, we train a CURL agent for 60K environment interactions on Walker-stand; then, we only fine-tune the policy and Q-functions on unseen tasks using network parameters from Walker-stand. To save computation, during fine-tuning, we freeze the encoder parameters. Figure 7a shows the computational gain of SEER in task transfer (i.e., Walker-stand to Walker-walk similar to Yarats et al. [52]), and domain transfer (i.e., Walker-stand to Hopper-hop) is shown in Figure 7b. Due to the generality of

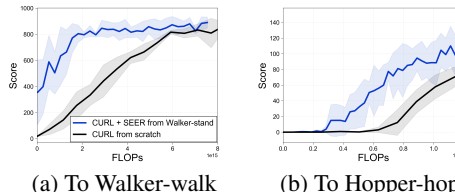

(a) To Walker-walk  (b) To Hopper-hop

Figure 7: Comparison of the computational efficiency of agents trained from scratch with CURL and agents trained with CURL+SEER from Walker-stand pretraining. The solid line and shaded regions represent the mean and standard deviation, respectively, across three runs.

CNN features, we can achieve this computational gain using a pretrained encoder. For the task transfer setup, we provide more analysis on the number of frozen layers and freezing time hyperparameter $T_f$ in Appendix **??**. While these transfer learning experiments are relatively independent to the compute-efficiency experiments in Section 5.2, we believe this is an exciting additional application of SEER and that more comprehensive investigations in this direction would be interesting future work.

## 5.5 Encoder analysis

In this subsection we present visualizations to verify that the neural networks employed in deep reinforcement learning indeed converge *from the bottom up*, similar to those used in supervised learning [36]. Figure 8a shows the spatial attention map for two Atari games and one DMControl environment at various points during training. Similar to Laskin et al. [24] and Zagoruyko & Komodakis [54], we compute the spatial attention map by mean-pooling the absolute values of the activations along the channel dimension and follow with a 2-dimensional spatial softmax. The attention map shows significant change in the first 20% of training, and remains relatively unchanged thereafter, suggesting that the encoder converges to its final representations early in training. Figure 8b shows the SVCCA [36] score, a measure of neural network layer similarity, between a layer and itself at time $t$ and $t + 10K$. The convolutional layers of the encoder achieve high similarity scores with themselves between time $t$ and $t + 10K$, while the higher layers of the policy and Q-network continue to change throughout training. In our DMControl environments we freeze the convolutional layers and the first fully-connected layer of the policy and Q-network (denoted fc1). Although the

policy fc1 continues to change, the convergence of the Q-network fc1 and the encoder layers allow us to achieve our computational and memory savings with minimal performance degradation.

We remark that while the encoder can be frozen early in RL training, using a randomly initialized encoder is ineffective [41]. It is important to train encoders on the task in order to learn useful features (as is done by widely used methods such as Srinivas et al. [39] and Laskin et al. [24]), but our finding is that these encoders converge early in task-specific training.

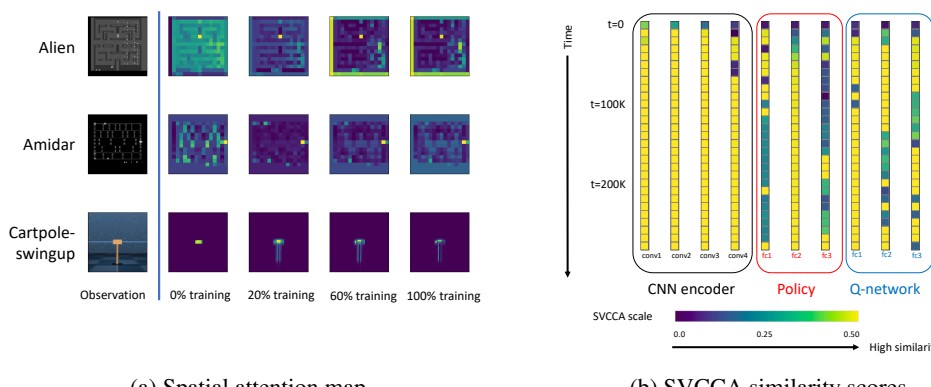

(a) Spatial attention map  (b) SVCCA similarity scores

Figure 8: Visualizations of encoder features throughout training. (a) Spatial attention map from CNN encoders. (b) SVCCA [36] similarity scores between each layer and itself at time $t$ and $t + 10K$ throughout training for Walker-walk task.

## 6 Discussion and Limitations

In this paper, we proposed a technique that reduces computation requirements for visual reinforcement learning, which we hope serves to facilitate a shift toward more compute-efficient RL. Here, we highlight other techniques for reducing training time. For experimentation in computationally intensive environments, Obando-Ceron & Castro [32] propose to use small- and medium-scale experiments, which could reproduce the conclusions of the Rainbow DQN paper in Atari games. For faster training time in a particular experiment, one can also lower the resolution of the input images. In Figures 9a and 9b we show that reducing the resolution by a factor of 2, from $100 \times 100$ to $50 \times 50$ (and scaling crops appropriately) produces significant compute-efficiency gain in DeepMind Control Suite without sacrificing performance, and emphasize that this technique can be combined with SEER for further improved efficiency. We remark that the additional gain from SEER is larger in more complex environments (e.g., Walker) where learning requires more steps. However, we find that naive resolution reduction may not generally be applicable across environments and may require domain knowledge in order to prevent excessive information loss. In Figures 9c and 9d we show that resolution reduction by a factor of 2, from $84 \times 84$ to $42 \times 42$, results in noticeably worse performance in several Atari games. In contrast, SEER successfully improves compute-efficiency without sacrificing performance in these games (see Figure 3). Overall, SEER is highly generalizable across visual domains, and can be easily combined with other modifications.

A limitation of our work is the introduction of a hyperparameter for the freezing time $t$. While domain knowledge can be used to decide a reasonable range for $t$ and reduce the search space, an interesting future direction would be to adaptively determine the freezing time using a metric of convergence. We also do not show the application of SEER to tasks which are more computationally expensive or even infeasible. We evaluate our method in DM Control and Atari because they are common RL benchmarks used in many recent works on RL from pixels, but the full impact of SEER may be more easily seen in very visually complex and challenging tasks such as 3D navigation. We do not foresee any negative societal impacts of our work, as it simply reduces training time of already existing algorithms.

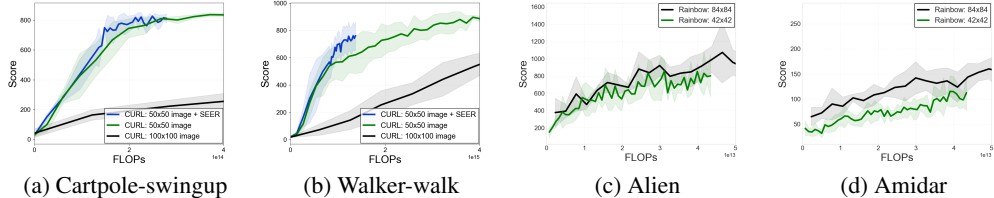

|                       |                       |                       |                       |
|-----------------------|-----------------------|-----------------------|-----------------------|
| (a) Cartpole-swingup  | (b) Walker-walk       | (c) Alien             | (d) Amidar            |

Figure 9: Evaluation of the compute-efficiency of CURL ((a) and (b)) and Rainbow ((c) and (d)) with original and reduced (by factor of 2) resolutions. The solid line and shaded regions represent the mean and standard deviation, respectively, across five runs.

## 7 Conclusion

We presented SEER, a simple but powerful modification of off-policy RL algorithms that significantly reduces computation and memory requirements while maintaining state-of-the-art performance. We leveraged the intuition that CNN encoders in deep RL converge to their final representations early in training to freeze the encoder and subsequently store latent vectors to save computation and memory. In our experimental results, we demonstrated the compute- and memory-efficiency of SEER in various DMControl environments and Atari games, and proposed a technique for compute-efficient transfer learning. With SEER, we highlight the potential for improvements in compute- and memory-efficiency in deep RL that can be made without sacrificing performance, in hopes of making deep RL more practical and accessible in the real world.

## 8 Acknowledgements

This research is supported in part by Open Philanthropy, ONR PECASE N000141612723, NSF NRI #2024675, and Berkeley Deep Drive. We would like to thank Kourosh Hakhamaneshi, Fangchen Liu, and anonymous reviewers for providing helpful feedback and suggestions. We would also like to thank Denis Yarats for the IMPALA encoder architecture implementation and Kai Arulkumaran for help with modifying the Rainbow DQN codebase.

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
