# OpenReview forum: "Improving Computational Efficiency in Visual Reinforcement Learning via Stored Embeddings"
_NeurIPS.cc/2021/Conference — NeurIPS 2021 Poster_

### Official Review · Reviewer_1FST · 2021-06-25

**Rating:** 6
**Confidence:** 5

**Summary:**

The paper propose a compute and memory efficient pixel wise RL algorithm by early stoping the update of CNN encoder and store the latent vector to the replay buffer instead of the images. Experiments had been done on DM Control and Atari with baseline algorithm CURL.


**Main Review:**

Strengths,

The paper is well organized and easy to follow.
The code, data, and instructions needed to reproduce the main experimental results is provided.
The compute cost is lower than baselines.
The memory cost is smaller.


Concerns and Questions,

1. The computation and the memory cost is not the very important issue in RL. The most concerns are the sample efficiency, since interact with the environment usually cost benefit loss or even damage accidents during the exploration. So although the result showed with FLOPs significantly outperformed the baseline, the result is not quit surprising, since the SEER no longer needs to compute the gradient of the encoder after freeze time. And the memory is not the concerns also, since the size of replay buffer is not the larger the better as discussed in “A Deeper Look at Experience Replay” and some other researches.

2. Missed some related works that improves the sample efficiency in pixels state reinforcement learning. To name a few, PISAC, DRQ, SLAC, the performances of them are better than CURL. I was wondering whether freeze it at a previous time harms the performance of those algorithms. Even with CURL, as shown in Fig. 12, before the freeze line the performance is quite similar, but the performance drops after the freeze time in most task.

3. It is hard to choice the time to freeze the encoder in practice. Seems like it needs more repetitions of the experiments to give a good choice of the freeze time? More experiments show the sensitive to the time of freeze of the algorithm would be better.



Suggestions and Discussions,
1. Before the encoder is freeze, the replay buffer still needs to store the images and if the replay buffer slots number is equal with the baseline settings, the performance is equal to the baseline in theory. After the freeze time, the replay buffer slots number grows more larger. So the benefit is seems like on the condition of the assumption that a larger replay buffer would brings performance improvement? Such assumption needs to be claimed and discussed more clearly in the paper. Further discussions and experiments on the different limitations of the memory cost would be helpful.

2. The part of transfer learning (Sec 5.4) is not quit clear to me. It seems like transfer learning is kind of a relatively independent experiment of what want to claim in this paper. If the paper want to claim that the SEER can bring benefit to transfer learning, more experiments has to be done to justify such benefit. If not, I think the paper without transfer learning experiment is enough for what the paper want to propose.



**Time Spent Reviewing:**

3

---

> ### Author Response · Authors · 2021-08-10
> **Response to Reviewer 1FST**
>
> We sincerely thank you for your helpful feedback and insightful comments. We appreciate that our paper is recognized for being well-organized, easy to follow, and reproducible, as well as reducing compute- and memory costs. We address your comments and questions below:
>
> ---
> **Q1. Computation and memory cost is not important in RL**
>
> **A1.** While sample-efficiency is important, there are many real-world scenarios where compute and memory are important too, such as training on small devices (e.g., on the scale of mobile phones, drones, Raspberry Pi's). Even in large-scale systems, compute and memory are not infinite, and SEER could be useful for training. Another potential benefit of reduced memory requirements is the ability to store the replay buffer in GPU and reduce expensive CPU to GPU transfers, allowing for fast data reads. In general, as Reviewer KTu3 mentioned, SEER allows researchers to have a faster iteration cycle, which is useful in many situations. There have recently emerged various techniques to reduce training time, such as that of Ceron et al. 2020, so the RL community indeed is concerned about compute-efficiency in addition to sample-efficiency.
>
> **Q2. Does freezing harm performance of other algorithms (e.g, PISAC, DrQ, SLAC)**
>
> **A2.** We ran initial experiments with RAD + SEER and DrQ + SEER, and found that the trends are similar to CURL + SEER. The performance degradation was similarly small and the compute gains were of a similar scale. We will include some RAD and DrQ results in the camera-ready version. There is no fundamental reason why SEER cannot be applied to other pixel-based RL algorithms!
>
> **Q3. Sensitivity to freezing time**
>
> **A3.** We will add these ablations to the camera-ready version! The general trend is that freezing time around $T=100K$ usually works well in Atari, and in our experiments, $T \in \\{50K, 100K, 150K\\}$ produce similar results so it is not particularly sensitive to freezing time. In DM Control you need to do per-environment hyperparameter tuning since the tasks are more varied.
>
> Edit: Please see https://sites.google.com/view/seer-freezing-time for learning curves for $T \in \\{50K, 100K, 150K\\}$. We will add these to the final draft. Thanks again helping us to improve our paper!
>
> **Q4. Assumption that larger buffer improves performance needs to be claimed and discussed more clearly**
>
> **A4.** We will add this assumption and relevant discussions/experiments to the paper. There are many real-world scenarios where memory may be limited, so increasing the capacity of the replay buffer is important (see A1 above)!
>
> **Q5. Transfer learning is kind of relatively independent experiment**
>
> **A5.** We agree that the paper without transfer learning is enough for what we want to propose. We wanted simply to highlight additional possibilities of our technique, and we will reframe the section on transfer learning to be more of a discussion section.
>
> References:
> - [Ceron et al. 2020]: Ceron, J.S.O. and Castro, P.S., 2021, July. Revisiting Rainbow: Promoting more insightful and inclusive deep reinforcement learning research. In International Conference on Machine Learning (pp. 1373-1383). PMLR.

---

### Official Review · Reviewer_qa64 · 2021-07-16

**Rating:** 6
**Confidence:** 4

**Summary:**

Stored Embeddings for Efficient Reinforcement learning (SEER) is a method to reduce the computational cost of value-based off-policy reinforcement learning, e.g. Q-learning, in settings with pixel-based observations. It uses a convolutional encoder whose lower layers are frozen early in agent training, as there has been previous work showing that a convolutional network's lower layers converge earlier on in training. Following an early period of joint training with the convolutional encoder, the agent afterwards stores only the encoded image features into its replay buffer instead of the raw pixels of the image. This enables potentially significant memory and compute savings as the image no longer needs to be re-encoded during off-policy learning. The paper presents empirical results demonstrating that final agent performance is not hindered by this procedure and that, when counting in terms of total FLOPs calculated or replay buffer size, SEER has an advantage over the naive image-based replay buffers most commonly used. A variety of experimental settings are considered, differing across domains (DMControl, Atari) and training procedure (Rainbow, SAC, CURL).

**Limitations And Societal Impact:**

The authors discuss some limitations of their method in section 6, mentioning the additional freezing time hyperparameter introduced by SEER.

There does not seem to be a potential for negative societal impact for this work.

**Main Review:**

The proposed method, SEER, can be seen as an application of Freezeout to value-based RL, combined with storing embeddings instead of images in the replay buffer after the lower layers are frozen. The method seems well-motivated and straightforward, leveraging the fact that the convolutional encoder trains quickly to avoid the need to re-encode images several times during learning. Considering that reinforcement learning is a considerably different setting to supervised learning, in that the input data distribution can change significantly over the lifetime of the agent, it was not immediately obvious that Freezeout can be applied without issues in this setting.

However, the authors have provided experimental results on pixel-based DMControl and Atari that seem to validate that freezing the lower convolutional layers early-on in training does not cause decreased performance in those domains. But there are still some remaining concerns I have regarding the generality of the proposed method that result in my score of weak reject:

1) A lot of the environments the method was evaluated on are highly regular in visual appearance, even throughout policy performance levels. I checked videos of the various Atari 2600 games, and only Krull has non-static backgrounds and a larger variety of visual representations for entities. The DMControl tasks are also very static across agent performance levels. I think it is important to test how this method would fare when there exists significant distribution shifts during training, e.g. an important new entity or visual mechanic appears at a certain policy performance. Would freezing the network before this critical performance level still make later policy learning improve?
2) Even not considering shifts in distribution, the visual understanding necessary for high performance in the environments tested is limited. As mentioned in the paper's section 6, it would be interesting to test SEER on more complex visual environments which require larger image encoders to extract useful semantics.
3) Maybe owing to the visual simplicity of the environments, the ablations reveal that the convolutional layers can be frozen early with minimal reduction in final performance. Have the authors tried using features from a randomly initialized network as a baseline.
4) What is the effect on wall-clock time of training an agent using SEER? What would the graphs look like if the x-axis was experiment run-time and all agents were trained on the same reference machine?

==========================================================================

Thanks for the response, the experimental results you described have addressed most of the concerns I had above. I think those results should be included in the camera-ready paper as an appendix section. I will raise my score to a 6.


**Time Spent Reviewing:**

4

---

> ### Author Response · Authors · 2021-08-10
> **Response to Reviewer qa64**
>
> We sincerely thank you for your helpful feedback and insightful comments. We appreciate that our paper is recognized for being well-motivated and straightforward. We address your comments and questions below:
>
> ---
> **Q1. How would this method fare when there exists significant distribution shifts**
>
> **A1.** We agree that it would be challenging to use SEER if the visual appearance of the game changes drastically, so this method should be used under the assumption that the changes are not severe. This should be a reasonable assumption in many scenarios, such as robotic manipulation and locomotion. However, SEER is indeed robust to some significant visual distribution shifts, even including domain transfer. In Figure 7b, we show that we can use a *frozen* encoder pretrained on Walker-stand for Hopper-hop training. The images encountered in these two tasks are very different, but the frozen encoder can successfully use those features in a new domain.
>
> **Q2. More complex visual environments**
>
> **A2.** We agree it would be interesting to test SEER on more complex visual environments, but also believe the current results sufficiently show its effectiveness, as DM Control and Atari are very challenging benchmarks for visual RL. Additionally, Figure 6 also shows that SEER is compatible with larger image encoders such as the IMPALA encoder. We did some initial experiments in Meta-World with DrQ and found SEER + DrQ to improve compute-efficiency while not causing significant performance degradation compared to DrQ. We aim to include these in the camera-ready version, but again, do not think the effectiveness of SEER is dependent on these results.
>
> **Q3. Features from a randomly initialized network as a baseline**
>
> **A3.** We have indeed run experiments with a randomly initialized convolutional encoder as a baseline, and it does not work well. Figure 5 of Stooke et al. 2020 also shows similar results. It is important to train encoders on the task in order to learn useful features (which is why methods such as CURL, RAD, and DrQ are widely used), but our finding is that these encoders converge early in task-specific training.
>
> **Q4. Effect on wall-clock time**
>
> **A4.** Given our computational constraints, it is difficult to accurately measure wall-clock time and we did not run all agents on the same machine without other jobs running. However, here is our best attempt at estimating some wall-clock times based on our logs:
>
> - In Finger-spin (DM Control), 400K environment steps with CURL takes 12 hours, CURL + SEER takes 8 hours
> - In Alien (Atari), 500K environment steps with Rainbow takes 8 hours, Rainbow + SEER takes 5 hours
>
> Given these numbers, we believe the graphs would look similar if the x-axis was wall-clock time. We will include wall-clock time graphs in the camera-ready version, with all agents trained on the same machine.
>
> References:
> - [Stooke et al. 2020]: Stooke, A., Lee, K., Abbeel, P. and Laskin, M., 2021, July. Decoupling representation learning from reinforcement learning. In International Conference on Machine Learning (pp. 9870-9879). PMLR.

---

### Official Review · Reviewer_KTu3 · 2021-07-16

**Rating:** 7
**Confidence:** 4

**Summary:**

- This paper presents a method for compute- and memory-efficient reinforcement learning where the visual encoder is frozen partway into training.  After freezing, latent vectors are stored in the replay buffer instead of images (and any existing images are replaced by them).  This leads to both better compute and memory utilization.
- The authors demonstrate their method by comparing to Rainbow on Atari and CURL on DM Control.  On DM control, their method reduces the compute by a considerable margin.  On Atari, the results are less clear cut, but the compute cost is reduced.
- When they also impose a memory constraint, the effectiveness of their method is further increased.


**Limitations And Societal Impact:**

Yes

**Main Review:**

- Strength
    - Elegant and "obvious in hindsight" (a good thing) idea, meaning it will likely have broad applicability
    - While the authors only tested it on off-policy methods, it is clearly also applicable to on-policy methods that use a rollout storage (PPO, PPG, IMPALA, V-MPO, A2C, etc.)
    - Good FLOPs vs. Reward results on DM Control
    - Results are more influential with a deeper CNN
- Weaknesses
    - The memory constrained results seem very contrived.  60 MB is a tiny amount of memory and even 9.0 GB of (presumably) CPU memory isn't that prohibitive.
        - Perhaps if wall-clock time was plotted in addition to samples, the smaller memory footprint of SEER would mean the replay buffer can be stored on the GPU and training would be much faster since many expensive CPU -> GPU transfers would be eliminated?
    - The CNN is frozen all at once instead of frozen iteratively. Raghu 2017 and Figure 6c suggest that the early layers could be frozen much earlier, although this may increase the memory usage initially since CNNs typically increase the memory size of the feature map in lower layers.
- Overall
    - This is a useful idea that improves experiment time, giving researchers a faster iteration cycle.  The idea is quite generic so it should be easy for researchers to adopt.
- References
    - Smith 2017: https://arxiv.org/abs/1711.00489


**Time Spent Reviewing:**

1

---

> ### Author Response · Authors · 2021-08-10
> **Response to Reviewer KTu3**
>
> We sincerely thank you for your helpful feedback and insightful comments. We appreciate that our paper is recognized for several positive aspects: (1) elegance and obviousness in hindsight, (2) applicability to off-policy and on-policy methods, (3) good compute-efficiency results, and (4) usefulness in research. We address your comments and questions below:
>
> ---
> **Q1: Memory constraints aren't prohibitive**
>
> **A1:** In use-cases like mobile robots where learning is asynchronous and completely on-device, 9GB of CPU RAM can be prohibitive or consume significant amounts of battery power. If the user wants to train for longer, the amount of memory required is even larger. We agree that another potential benefit is the ability to store the replay buffer in GPU and reduce expensive CPU to GPU transfers and think this would be very interesting future work!
>
> **Q2: Freezing layers iteratively**
>
> **A2:** In deep RL, the convolutional networks are usually shallower than those used in supervised learning and other domains, so we find that freezing the encoder layers all at once provides a considerable compute gain already. We did run initial experiments with iterative freezing and did not find a significant gain over freezing all at once.

---

### Author Response · Authors · 2021-08-26
**A gentle reminder**

Dear Reviewers,

Thank you for your time and efforts in reviewing our paper.

We gently remind you that we are reaching the end of the discussion period. We believe that we sincerely and successfully addressed your concerns/questions/suggestions. If you have any further concerns or questions, please do not hesitate to let us know.

Thank you very much! - Authors

---

### Decision · Program_Chairs · 2021-09-27

**Decision:**

Accept (Poster)

**Comment:**

An interesting approach to addressing the problem of dealing with vision input in the reinforcement learning setting. Presented approach is more computationally efficient than its counterparts. The algorithm is also flexible enough and can be applied in the on-policy context, even though in the paper the authors focus on the off-policy setting. Well written paper. All the reviewers see the importance of the proposed method in dealing with the notoriously difficult problem of computationally-efficient vision-based RL.